# Development of an Anti-Idiotypic VHH Antibody and Toxin-Free Enzyme Immunoassay for Ochratoxin A in Cereals

**DOI:** 10.3390/toxins11050280

**Published:** 2019-05-20

**Authors:** Caixia Zhang, Qi Zhang, Xiaoqian Tang, Wen Zhang, Peiwu Li

**Affiliations:** 1School of Life Sciences, Hubei University, Wuhan 430062, China; zcx1778253705@163.com (C.Z.); zhangqi01@caas.cn (Q.Z.); wtxqtutu@163.com (X.T.); zhangwen@oilcrop.cn (W.Z.); 2Oil Crops Research Institute of the Chinese Academy of Agricultural Sciences, Wuhan 430062, China; 3Key Laboratory of Biology and Genetic Improvement of Oil Crops, Ministry of Agriculture, Wuhan 430062, China; 4Key Laboratory of Detection for Mycotoxins, Ministry of Agriculture, Wuhan 430062, China

**Keywords:** anti-idiotypic nanobody, surrogate, Ochratoxin A, enzyme-linked immunosorbent assay

## Abstract

Enzyme-linked immunosorbent assay (ELISA) test kits have been widely used for the determination of mycotoxins in agricultural products and foods, however, this test uses toxin standards with high toxicity and carcinogenicity that seriously threaten human health. In this work, the anti-idiotypic nanobody VHH 2-24 was first developed and then, using it as a surrogate standard, a toxin-free enzyme immunoassay for ochratoxin A (OTA) was established. The IC_50_ value of the VHH 2-24 surrogate standard-based ELISA was 0.097 µg/mL, with a linear range of 0.027–0.653 µg/mL. The average recoveries were tested by spike-and-recovery experiments, and ranged from 81.8% to 105.0%. The accuracy of the developed ELISA for detecting OTA was further verified by using the high performance liquid chromatography (HPLC) method, and an excellent correlation was observed. In summary, the toxin-free ELISA established in this study proves the latent use of the anti-idiotypic VHH as a surrogate calibrator for other mycotoxins and highly toxic small molecule analysis to improve assay properties for highly sensitive analyte determination in agricultural products.

## 1. Introduction

Ochratoxin A (OTA) is a secondary metabolite produced by the genus Aspergillus and Penicillium, especially A. ochraceus and P. verrucosum [1,2]. OTA mainly contaminates cereals, vegetables, coffee beans, fruits, cocoa and their derivatives such as bread, coffee and wine [3,4]. In addition, many studies have shown that OTA has a variety of toxic effects, including nephrotoxicity, hepatotoxicity, immunosuppression, carcinogenicity and teratogenicity [5,6,7], which has potential harm to animal and human [8]. In humans, the International Agency for Research on Cancer (IARC) has categorized ochratoxin A as a possible carcinogen (group 2B) [9]. Due to its widespread presence and chemical stability in food [10,11], European countries have established strict regulatory limits for OTA in order to reduce public health risks [12]. In addition, there is an urgent need to establish low-cost, rapid, and highly sensitive methods to detect the presence of OTA in food to reduce its harm.

To date, a variety of analytical methods have been developed for the determination of OTA, including the thin-layer chromatographic (TLC) method, high-performance liquid chromatography, immunoassay, and electrochemical sensor methods [13,14,15,16]. Instrumental analysis methods are precise and sensitive; nevertheless, those methods are time-consuming and require complex sample pre-processing and expensive equipment. Alternatively, immunoassays based on the specific binding of the antigens and antibodies have the advantages of rapid sample preparation, selectivity, high sensitivity and cost effectiveness [17,18]. This means they are suitable for rapid screening of numerous samples in food safety testing. OTA is a small molecule compound; hence it is necessary to use a competitive immunoassay format to detect it [19]. However, in the establishment and application of immunoassay methods, the use of a large number of coating antigens synthesized by organic solvents and OTA standards seriously threaten the health of operators [20]. In addition, the waste liquid generated during the experiment may also cause secondary pollution of the surrounding environment. Therefore, we urgently need to develop a toxic-free immunoassay for detecting OTA.

In recent years, numerous studies have reported the superiority of developing toxin substitutes for immunoassays, mainly involving aflatoxins, deoxynivalenol, ochratoxin A, fumonisins, zearalenone and citrinin [21,22,23,24,25,26]. These alternatives can be obtained by the preparation of anti-idiotypic antibodies. In 1974, Jerne [27] first proposed the “network theory of the immune system”, the core of which is the idiotype (Id) of antibodies that can induce the body to produce the anti-idiotypic (Anti-id or AId) antibodies. Anti-id antibodies, which are capable of specifically binding to the antigen-binding region of the antibody, form a relationship of “internal images” with the antigen, mimic the three-dimensional structure of the antigen, and thus act as a substitute for the antigen molecule [28,29]. According to the literature, monoclonal and polyclonal anti-id antibody technologies have been widely applied in vaccine development [30], disease treatment [31] and small molecule detection [32]. The anti-id polyclonal antibodies can be easily obtained in a short period of time at a low production cost; however, the single yield of antibodies is limited and there is variability in the antibodies from different animals. Although the monoclonal antibody can produce a stable single anti-id antibody, it has a longer production cycle and is an expensive process. Modern genetic engineering technology has provided new tools for the development of anti-id antibodies, such as fragment antigen binding (Fab) [33] and single chain variable fragments (scFv) [34]. Nevertheless, the shortcomings of Fab and scFv—low solubility and stability—prevent their wide application. In 1993, scientists discovered that there is a type of antibody that naturally lacks a light chain in the camelids [35] and sharks [36], which is called a heavy chain antibody. A single-domain heavy chain antibody containing only the heavy chain variable region can be produced by cloning and recombinant expression of the variable region of the heavy chain antibody. The Belgian company, Ablynx calls it a nanobody or VHH because its molecular weight is only about 17KD [37]. Compared with traditional recombinant antibodies, nanobodies have better solubility, high thermal stability and high tolerance to organic solvents [38]. In view of these advantages, VHH antibodies are more suitable for the development of alternative reagents for immunoassay. The anti-id nanobody is used as a surrogate for antigens and standards have been applied to the toxic-free green immunoassays for aflatoxins [21,39], further prompting us to study its use as an alternative to other mycotoxins. However, there are no reports on a toxin-free competitive ELISA for analyzing OTA by using anti-id VHH antibody as a surrogate standard.

In this study, we report the specificity and thermal stability of the anti-id nanobody VHH 2-24 developed in our experiment. The obtained VHH 2-24 was used as the surrogate standard to develop a toxic-free indirect competitive enzyme-linked immunosorbent assay for the detection of OTA in agricultural products.

## 2. Results and Discussion

### 2.1. Expression and Purification of the Anti-Id Nanobody

TOP10F’ is a non-suppressor cell that permits the expression of dissolvable VHH without the pIII protein. For expression, the phagemid containing the VHH gene was transformed into *E. coli* Top10F’, and the *E.coli* ER2738 periplasmic protein was extracted with xTractor buffer. The VHH 2-24 containing 6 × His tag was purified on the Ni−NTA resin column. Then, the purified VHH 2-24 was characterized by a 12% SDS-PAGE gel. As shown in Figure 1, the size of the obtained VHH 2-24 is about 15 kDa, which is in accordance with the results computed by the protein information resource. There is only one band of target protein in the figure, which proves that the purification effect of VHH is good. We measured the concentration of the purified VHH 2-24 by the Bradford method and the result is 211.2 µg/mL.

### 2.2. Specificity of the Anti-Id Nanobody

In our previous work, we demonstrated that the nanobody VHH 2-24 could be used as a coating antigen, which was shown to specifically bind to the anti-OTA mAb 1H2 (Figure 2), thus, we inferred that the VHH might specifically recognize the antigen binding site of the antibody. In order to prove this, we conducted a verification test. In this study, the specificity of the anti-id nanobody VHH 2-24 was determined by competitive ELISA with three mAbs: anti-AFB1 mAb, anti-DON mAb and anti-ZEN mAb. No conspicuous inhibition was observed when various concentrations of VHH 2-24 were mixed with three mAbs; however, there was a significant inhibition of the binding between OTA and anti-OTA mAb 1H2. Therefore, these results showed that VHH 2-24 could be highly selective and could specifically bind with the variable region of the mAb 1H2.

### 2.3. Thermal Stability of the VHH Surrogate Calibrator

The thermal stability of the VHH antibody has played an important role in improving product stability and service life when VHH is used as an immunoassay reagent. First, we performed the thermal stability test on VHH 2-24 at different temperatures to investigate this issue. The VHH 2-24 solution diluted to a working concentration with PBS buffer was heated for 5 min at 20 °C, 37 °C, 50 °C, 60 °C, 70 °C, 80 °C and 90 °C, respectively. After cooling to room temperature, the binding ability of the treated VHH 2-24 with the monoclonal antibody 1H2 was tested by indirect non-competitive ELISA. As shown in Figure 3(a), we observed that as the temperature increases, the mAb 1H2 reactivity with OTA-BSA gradually decreases while the binding capacity of the VHH 2-24 hardly changes and VHH can still bind to the antibody at a temperature of 90 °C. 

To further verify this, the thermostability of VHH 2-24 was also studied by comparison with the monoclonal antibody 1H2 at various incubation times. VHH 2-24 and the monoclonal antibody 1H2 were incubated to 80 °C for different times (0, 5, 10, 20, 30, 40, 50, 60 min). Each of the samples was re-equilibrated to RT, followed by assaying for their binding activity. From Figure 3b, it can be seen that the monoclonal antibody 1H2 immediately lost its binding ability after incubation at 80 °C for 10 min. However, VHH 2-24 retained about 50% of its binding capacity after heating for 40 min at 80 °C. Consequently, VHH 2-24 has better thermostability than conventional antibodies and is more suitable for the development of alternative reagents for immunoassays. This result was expected because VHH can form an additional disulfide bond between CDR3 and CDR1 or FR2 in addition to the conserved disulfide bond within the domain. Hence, the increased sequence and loop structure of VHH expands the area of antibody binding to antigens and the diversity of antibodies while increasing the stability of its structure, resulting in VHH being able to withstand high temperatures and harsh extreme environments.

### 2.4. Correlation between Anti-Id VHH and OTA

In order to compare the OTA standard solutions with the VHH 2-24 solutions, standard inhibition curves were established by indirect competitive ELISA. The OTA standard curve (Figure 4a), which was extremely similar to the curve using the VHH as standard (Figure 4b), was S-shaped and suitable for the four-parameter logic equation. It was clear that the two standard curves had a linear range between 20% and 80% of the binding rate, indicating their obvious inhibition effect. To evaluate whether the anti-id VHH 2-24 could supersede OTA as a standard substance, we analyzed the correlation between the concentrations of VHH 2-24 and OTA. Based on the above two standard curves, we calculated the concentrations of OTA and VHH 2-24 corresponding to the same binding rate between 20–80% and established a linear relationship. From Figure 5, the concentration of the OTA and VHH 2-24 standards material exhibited a good linear relationship (*y* = 1.0973*x* − 0.0639, *R*^2^ = 0.9854). In brief, the results proved that VHH 2-24 could be used as an OTA surrogate calibrator for indirect competition ELISA.

### 2.5. Surrogate Standard Curve

With a two-step calculation, OTA in actual samples was detected by the ELISA method based on a VHH 2-24 surrogate standard. The calculation method was as follows: First, the B/B_0_ value of each well was obtained by the OD value of the samples measured by ELISA, and it was applied to the four-parameter logistic equation of the nanobody substitution to calculate the corresponding concentration of VHH 2-24. The four-parameter logic equation is as follows:(1)$$y=14.2118+\frac{85.7392}{1+{\left(\frac{x}{0.0735}\right)}^{1.2016}},\;R^{2}=0.9988$$
where y is the B/B_0_ value and *x* is the anti-id nanobody VHH 2-24 concentration.

Then, the concentration of VHH 2-24 was converted into the corresponding OTA concentration by a quantitative conversion equation of VHH 2-24 and OTA concentration as follows:(2)$$y=1.0973x-0.0639,\;R^{2}=0.9854\;$$
where y is the OTA concentration and *x* is the VHH 2-24 concentration.

The two-step calculation displayed an excellent detection limit of 0.001 ng/mL (IC_10_) and a wide linear range for the OTA (IC_20_–IC_80_) of 0.003–0.673 ng/mL. Hence, OTA could be analyzed by using a toxin-free ELISA based on a VHH as a surrogate material.

### 2.6. Assay Validation

To validate the accuracy of the toxin-free ELISA based-VHH for sample analysis, we performed spike-and-recovery experiments. Corn, rice and wheat samples spiked with serial concentrations of OTA standard solution (2 µg/kg, 10 µg/kg and 50 µg/kg) were made on the pretreatment. The OTA content in the sample extract was detected by the newly established VHH based ELISA. As seen in Table 1, the results showed excellent correlations between the spiked and recovered concentrations. For intra-assay, an average recovery of 81.8% to 105.0% was achieved; meanwhile, the recovery for the inter-assay ranged from 88.0% to 93.0%. Overall, the above assays indicate that the developed VHH 2-24 based-ELISA in this study was suitable for the determination of OTA in agricultural products.

### 2.7. Comparison between the ELISA Based on VHH and the HPLC Method

Ten corn samples naturally contaminated with OTA were selected and the levels of OTA were determined by the newly developed ELISA and HPLC methods. Linear regression analysis was performed on the OTA concentrations measured by the two methods. As shown in Figure 6, the results of the two methods showed a good linear relationship (*y* = 1.0383*x* − 0.3787, *R*^2^ = 0.994), which further proved that the detection of OTA in the sample by the toxin-free ELISA was accurate and reliable.

## 3. Conclusions

In this study, we developed a toxin-free, sensitive and selective ELISA for the analysis of OTA based on a VHH surrogate standard. We also demonstrated the particular benefit of the proposed immunoassay, that is, the method can effectively prevent the health risk to the operator caused by OTA during the testing process. By immunizing alpaca with the anti-OTA mAb 1H2, an anti-id nanobody VHH 2-24 for OTA was successfully obtained. As we have proved, VHH 2-24 showed fine specificity and improved thermal stability. What is more, this study analyzed the correlation between the anti-id VHH antibody and OTA and showed good linearity, with an R2 value of 0.9854. The accuracy of the newly developed method for OTA detection in spiked samples was verified, and the test results also revealed good recovery ranging from 81.8% to 105.0%. Comparing the results of the OTA corn sample with the reference HPLC method, we observed a good correlation (*y* = 0.81*x* + 9.82, *R*^2^ = 0.9922). In addition, the European Union has set strict maximum limits for OTA in food and feed, such as 10 µg/kg for instant coffee, 2 µg/kg for wine and 5 µg/kg for wheat. In conclusion, the above results indicate that the toxin-free ELISA established here is feasible for detecting OTA and it fulfills the requirements of the current legislation concerning the detection of OTA contamination in food and feed. Anti-id VHH has great potential in the development of immunoassay replacement reagents.

## 4. Materials and Methods 

### 4.1. Materials and Reagents

All inorganic chemicals and organic solvents were of reagent grade unless stated otherwise. Anti-ochratoxin A monoclonal antibody (mAb) 1H2 [19], anti-aflatoxin B1 mAb 1C11 [40], anti-zearalenone mAb 2D3 [41] and anti-deoxynivalenol mAb 1D5 were produced in our laboratory. Nanobody VHH 2-24 was obtained and purified by following our previous instructions. Ochratoxin A standard, bovine serum albumin (BSA), polyethylene glycol 8000(PEG8000), Freund’s incomplete adjuvant, isopropyl β-D-1-thiogalactopyranosi-de (IPTG), 3,3′,5,5′-tetramethylbenzidine (TMB) and goat anti-mouse monoclonal antibody conjugated to horseradish peroxidase (HRP) were procured from Sigma (St. Louis, MO, SA). The Costar 96-well EIA/RIA plate was purchased from Corning Incorporated (Corning, NY, USA). Tween 20 was obtained from the J&K Scientific (Beijing, China). XTractor buffer for protein extraction and His60 Superflow Resin were purchased from Clontech Laboratories, Inc. Mountain View, CA, USA). 0.01 M phosphate-buffered saline (PBS, pH 7.4) was prepared by adding 8 g of NaCl, 2.9 g of Na_2_HPO_4_·12H_2_O, 0.2 g of KH_2_PO_4_ and 0.2 g of KCl in 1000 mL deionized water.

### 4.2. Safety

Pure OTA standards were handled with care in the fume hood. All containers in contact with phage, OTA and bacterial cultures (centrifuge bottles, vials, glassware, tubes, etc.) were immersed in a 10% bleach solution for 2−3 h before processing and then autoclaved. All disposable items (centrifuge tubes and pipette tips) were autoclaved before they were discarded.

### 4.3. Expression and Purification of Anti-Id Nanobody to OTA

The pComb3X phagemid vectors encoding VHHs inhibited binding to the coating antigen and unique DNA sequences were extracted from ER2738 clones and transformed into the *E. coli* strain TOP10F′ cells. For expression, a single colony of TOP10F′ cell carrying nanobody expression plasmid was selected and incubated in 4 mL of SB overnight. Two ml of the overnight culture was added to a 500 mL cultural flask, which contained 200 mL of SB medium with ampicillin (50 μg/mL) and tetracycline (20 μg/mL), and this was shaken at 37 °C at 250 rpm. When the OD600 value of the culture reached about 0.6, IPTG was added into the culture with a final concentration of 1 mM and continuously shaken overnight at 250 rpm at 37 °C. 

The cell cultures were centrifuged at 8000× *g* for 10 min at 4 °C and periplasmic proteins were extracted by a XTractor buffer, then the VHH containing 6 × His tag was purified with Ni−NTA metal affinity chromatography according to the manufacturer’s instruction. The obtained soluble VHH was analyzed by SDS-PAGE for purity and size, and then concentrated by ultrafiltration tubes. Finally, the concentration of purified nanobody was determined by the Bradford method and stored at −20 °C.

### 4.4. Competitive ELISA Using OTA As Standard

The 96-well ELISA plate was coated with 100 μL/well of OTA-BSA (0.2 µg/mL) and incubated overnight at 4 °C. The next day, the ELISA plate was washed three times with PBST (PBS with 0.05% Tween 20, *v*/*v*), and then blocked with 300 µL/well of 3% skimmed milk in PBST for 1 h at 37 °C. After washing the plate three times again, 50 μL of each serial concentration of OTA standard diluted with 10% methanol/PBS (*v*/*v*) was mixed with an equal volume of monoclonal antibody 1H2 in PBS, and the mixture was added to each well of the microplate. After incubation for 1 h at 37 °C and washing for three cycles with PBST, 100 μL of goat anti-mouse antibody conjugated with HRP was added to each well followed by 1 h incubation at 37 °C. The color was developed by adding 100 μL peroxidase substrate (25 mL of 0.1 M citrate acetate buffer [pH 5.5], 0.4 mL of 6 mg/mL TMB in dimethyl sulfoxide (DMSO), and 0.1 mL of 1% H_2_O_2_). The plate was incubated for 15 min at 37 °C until enzyme reactivity was terminated by adding 50 μL of 2 M H_2_SO_4_. The absorbance was measured by a microplate reader (Molecular Devices, State of California, USA) at 450 nm.

### 4.5. Competitive ELISA Based on an Anti-Id VHH as Surrogate Standard

In addition to using continuous concentrations of nanobody in place of free OTA, we established the same assay as competitive ELISA using OTA as a competitor.

### 4.6. Sample Preparation

Samples of rice, corn, and wheat purchased from the local supermarket were finely ground using a grinder and stored in the freezer at −20 °C before use. Five grams of each milled sample was weighed separately and extracted using 80% methanol in ultrapure water (*v*/*v*). The mixture was shaken at 250 rpm at room temperature for 30 min, and then centrifuged at 5000 *g* for 30 min; the filtered supernatant was diluted four times with 4% BSA/PBS (*w*/*v*) and used for ELISA analysis after dilution.

### 4.7. HPLC-ELISA-Based Validation

To assess the accuracy of the newly established indirect competitive ELISA method based on VHH as surrogate standard, the ten randomly selected corn samples naturally contaminated by OTA were analyzed by ELISA and compared with the results of the HPLC method. OTA analysis in the sample extract was performed on a Shiseido C18 analytical column (particle size 3 µm; 150 × 4.6). The excitation and emission wavelengths of the fluorescence detector were set to 333 nm and 470 nm, respectively. The mobile phase consisted of a 45% methanol/water solution at a flow rate of 1 mL/min. The sample injection volume was 10 µL and the column temperature was 30 °C.

## Figures and Tables

**Figure 1 toxins-11-00280-f001:**
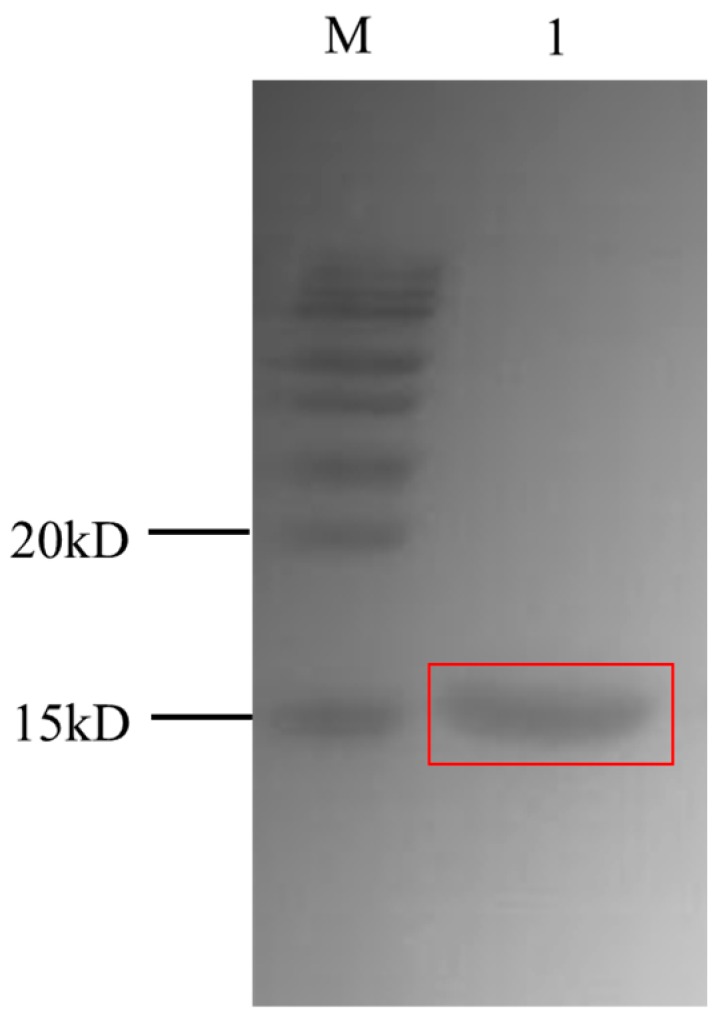
SDS-PAGE analysis of the purified VHH2-24 nanobody on 12% gel.

**Figure 2 toxins-11-00280-f002:**
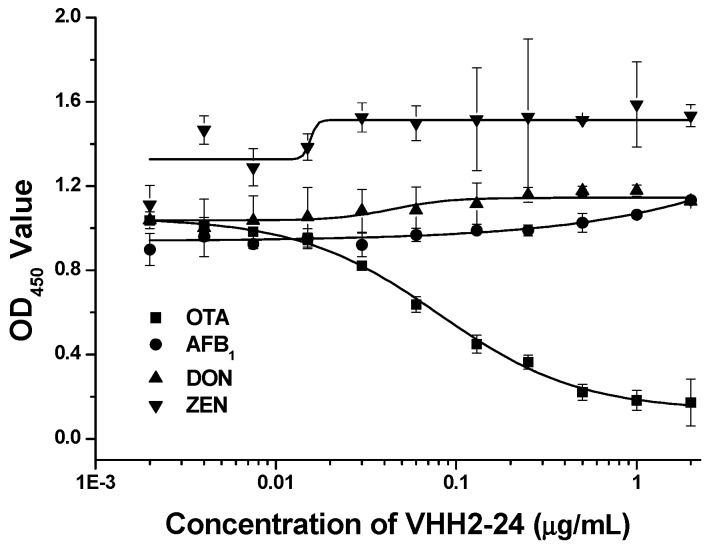
Specificity of VHH 2-24 towards anti-OTA, anti-AFB1, anti-DON and anti-ZEA monoclonal antibodies.

**Figure 3 toxins-11-00280-f003:**
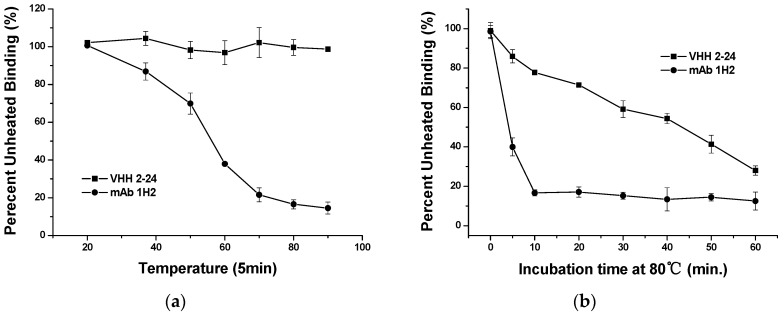
(**a**) Inhibition curves using VHH 2-24 as a standard surrogate after treatment under different temperatures; (**b**) Thermal stability of VHH 2-24 and monoclonal antibody 1H2.

**Figure 4 toxins-11-00280-f004:**
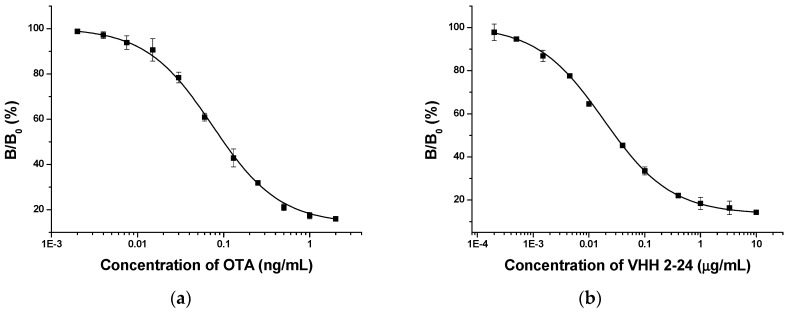
Standard curves of ELISAs using OTA (**a**) and VHH2-24 (**b**) as standards.

**Figure 5 toxins-11-00280-f005:**
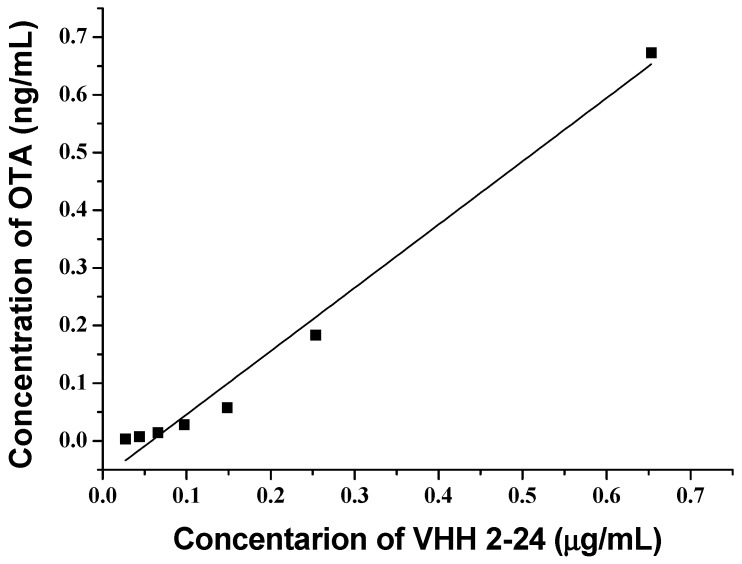
The linear relationship between the concentrations of OTA and VHH 2-24.

**Figure 6 toxins-11-00280-f006:**
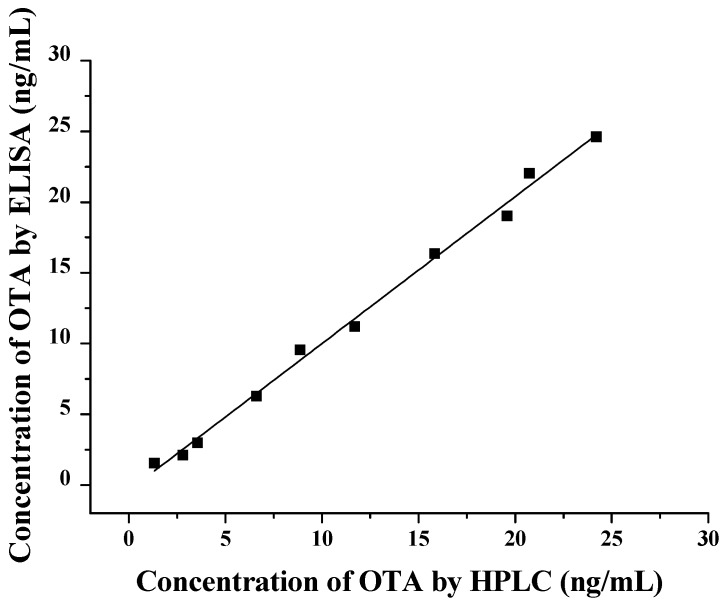
Correlation between the results of HPLC and ELISAs for OTA in contaminated corn samples.

**Table 1 toxins-11-00280-t001:** Recovery analysis of ELISA for OTA in agro-products.

Sample Types	Spike Level (µg/kg)	Means ± SD (µg/kg)	Recovery (%)
within assay (*n* = 3) ^a^			
Corn	50	41.2 ± 5.1	82.4
	10	8.7 ± 0.7	87.0
	2	1.8 ± 0.2	90.0
Rice	50	44.3 ± 5.6	88.6
	10	10.2 ± 0.4	102.0
	2	2.1 ± 0.3	105.0
Wheat	50	40.9 ± 4.9	81.8
	10	9.7 ± 1.0	97.0
	2	1.9 ± 0.4	95.0
between assay (*n* = 5) ^b^			
Corn	50	46.5 ± 6.7	93.0
	10	8.8 ± 0.7	88.0
	2	1.8 ± 0.2	90.0

^a^ Each assay was carried out in three replicates on the same day. ^b^ The assays were carried out on five different days.

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
