# Peer review of "Development of an Anti-Idiotypic VHH Antibody and Toxin-Free Enzyme Immunoassay for Ochratoxin A in Cereals"

_toxins, 2019, doi:10.3390/toxins11050280_

Round 1

Reviewer 1 Report

The manuscript “Development of an anti-idiotypic VHH antibody and toxin-free enzyme immunoassay for ochratoxin A in cereals” is clearly written and structured. The above mentioned topic is mainly covered from a laboratory-safety point of view.  From a diagnostic perspective it would be interesting for the reader to learn whether or not the assay is suitable to fulfill the requirements of the current legislation concerning the detection of ochratoxin A contamination in food and feed. An additional paragraph dealing with this aspect should be included either in the discussion or in the conclusion section.

Minor typewriting revisions

Line 59: include a blank after [31]

Lines 86 and 87: E. coli in italics and a blank after E. coli in line 87

Line 121: delete “The” at the beginning of sentence 2

Line 155: Please replace “wonderful” by a more objective description of the detection limit

General remarks to Fig. 2-6

Please include error bars in the figures. In Fig. 2; 3a and b a regression line instead of a line-and-dot plot is more suitable.

Author Response

Response to Reviewer 1 Comments

Point 1: The manuscript “Development of an anti-idiotypic VHH antibody and toxin-free enzyme immunoassay for ochratoxin A in cereals” is clearly written and structured. The above mentioned topic is mainly covered from a laboratory-safety point of view.  From a diagnostic perspective it would be interesting for the reader to learn whether or not the assay is suitable to fulfill the requirements of the current legislation concerning the detection of ochratoxin A contamination in food and feed. An additional paragraph dealing with this aspect should be included either in the discussion or in the conclusion section. 

Response 1: Thank you for your careful review. The sections of conclusion have been rewritten after carefully amendment, which have been revised to "In addition, the European Union has set strict maximum limits for OTA in food and feed, such as 10 µg/kg for instant coffee, 2 µg/kg for wine and 5 µg/kg for wheat. In conclusion, the above results indicated that the toxin-free ELISA established here is feasible to detect OTA and is suitable to fulfill the requirements of the current legislation concerning the detection of OTA contamination in food and feed, and anti-id VHH has great potential in the development of immunoassay replacement reagents.".

Point 2: Line 59: include a blank after [31]

Response 2: Thank you for your careful review. The sections of lines number 59 have been rewritten after carefully amendment, which have been revised to "disease treatment [31] and".

Point 3: Lines 86 and 87: E. coli in italics and a blank after E. coli in line 87

Response 3: Thank you for your careful review. The sections of lines number 86 to 87 have been rewritten after carefully amendment, which have been revised to "the phagemid containing the VHH gene was transformed into E. coli Top10F’, and the E.coli ER2738 periplasmic protein was extracted with xTractor buffer.".

Point 4: Line 121: delete “The” at the beginning of sentence 2

Response 4: Thank you for your careful review. The sections of lines number 121 have been rewritten after carefully amendment, which have been revised to "Each of the samples was re-equilibrated to RT,".

Point 5: Line 155: Please replace “wonderful” by a more objective description of the detection limit

Response 5: Thank you for your careful review. The sections of lines number 155 have been rewritten after carefully amendment, which have been revised to "The two-step calculation displayed an excellent detection limit of 0.001 ng/mL (IC10) and a wide linear range of the OTA (IC20-IC80) of 0.003-0.673 ng/mL.".

Point 6: General remarks to Fig. 2-6. Please include error bars in the figures. In Fig. 2; 3a and b a regression line instead of a line-and-dot plot is more suitable.

Response 6: Thank you for your careful review. After modification, the figures 2 to 4 have been included error bars. However, figure 5 is obtained from two standard curves, so there is no way to set the error line. According to previous research papers, the correlation curves of ELISA and HPLC results of mycotoxins in contaminated samples were not set with error bars. The figure 2 have been revised to a regression line. In addition, the figure 3 shows the change in the binding ability of the treated antibody to the antigen, and the results are not linear relationship, so regression analysis is not possible.

Reviewer 2 Report

The manuscript presents the data derived from the development of an anti-idiotypic VHH antibody and toxin-free enzyme immunoassay for ochratoxin A in cereals. The authors observed that the toxin-free ELISA established in their study is able to detect OTA in agricultural products and that anti-id VHH has great potential in the development of immunoassay replacement reagents. I recommend that this paper be accepted for publication after the following minor revisions

L9 change “…was firstly developed” to “…was first developed”  Line 27 – 28, note genus that produced OTA - Aspergillus and Penicillium especially Aspergillus ochraceus and Penicillium verrucosum. Change “…. metabolite that was produced by the genus Aspergillus ochraceus and Pecillium” to “…… metabolite produced by the genus Aspergillus and Penicillium especially A. ochraceus and P. verrucosum.

Line 31, delete “… great …”

Line 33, change “… chemical stability food” to “…chemical stability in food”

Line 34, change “…severe limit standard...” to “…strict regulatory limits…”

Line 36, Please restructure the sentence “Therefore, it is urgent…..” to correlate with the preceding sentence.

Line 38, delete “mainly”

Line 103, Please restructure the sentence for more clarity “…while there is a significantly inhibited to the ….”

Line 104, change “VHH 2-24 could highly” to “VHH 2-24 could be highly”

Line 113, VHH 2-24 andsolutions? Please correct accordingly

Line 121, Change “The each sample was…” to “ Each of the samples was…”

Line 185 - 186, change “…threaten caused by the OTA …” to “…risk caused by OTA …”

Line 255 correct “… HLPC method” to “…HPLC method”.

Author Response

Response to Reviewer 2 Comments

Point 1: Line 9 change “…was firstly developed” to “…was first developed”

Response 1: Thank you for your careful review. The sections of lines number 9 have been rewritten after carefully amendment, which have been revised to "the anti-idiotypic nanobody VHH 2-24 was first developed.".

Point 2: Line 27 – 28, note genus that produced OTA - Aspergillus and Penicillium especially Aspergillus ochraceus and Penicillium verrucosum. Change “…. metabolite that was produced by the genus Aspergillus ochraceus and Pecillium” to “…… metabolite produced by the genus Aspergillus and Penicillium especially A. ochraceus and P. verrucosum.

Response 2: Thank you for your careful review. The sections of lines number 27 to 28 have been rewritten after carefully amendment, which have been revised to "Ochratoxin A is a secondary metabolite produced by the genus Aspergillus and Penicillium especially A. ochraceus and P. verrucosum [1,2].".

Point 3: Line 31, delete “… great …”

Response 3: Thank you for your careful review. The sections of lines number 31 have been rewritten after carefully amendment, which have been revised to "which has potential harm to animal and human [8].".

Point 4: Line 33, change “… chemical stability food” to “…chemical stability in food”

Response 4: Thank you for your careful review. The sections of lines number 33 have been rewritten after carefully amendment, which have been revised to "Due to its widespread presence and chemical stability in food [10,11],".

Point 5: Line 34, change “…severe limit standard...” to “…strict regulatory limits…”

Response 5: Thank you for your careful review. The sections of lines number 34 have been rewritten after carefully amendment, which have been revised to  "European countries have established strict regulatory limits for OTA to reduce public health risks [12].".

Point 6: Line 36, Please restructure the sentence “Therefore, it is urgent…..” to correlate with the preceding sentence.

Response 6: Thank you for your careful review. The sections of lines number 36 have been rewritten after carefully amendment, which have been revised to  "In addition, it is urgent to establish the low-cost, rapid, and highly sensitive methods to detect the presence of OTA in food to reduce its harm.".

Point 7: Line 38, delete “mainly”

Response 7: Thank you for your careful review. The sections of lines number 38 have been rewritten after carefully amendment, which have been revised to  "To date, a variety of analytical methods have been developed for the determination of OTA, including thin-layer chromatographic (TLC) method,".

Point 8: Line 103, Please restructure the sentence for more clarity “…while there is a significantly inhibited to the ….”

Response 8: Thank you for your careful review. The sections of lines number 103 have been rewritten after carefully amendment, which have been revised to  "None of conspicuous inhibition were observed when various concentration of VHH 2-24 were mixed with three mAbs; howere, there is a significantly inhibited to the binding between OTA and anti-OTA mAb 1H2.".

Point 9: Line 104, change “VHH 2-24 could highly” to “VHH 2-24 could be highly”

Response 9: Thank you for your careful review. The sections of lines number 104 have been rewritten after carefully amendment, which have been revised to  "these results exhibited that VHH 2-24 could be highly selective and special binding with the variable region of the mAb 1H2.".

Point 10: Line 113, VHH 2-24 andsolutions? Please correct accordingly

Response 10: Thank you for your careful review. The sections of lines number 113 have been rewritten after carefully amendment, which have been revised to  "The VHH 2-24 solution diluted to a working concentration with PBS buffer was heated for 5 minutes at 20℃, 37℃, 50℃, 60℃, 70℃, 80℃and 90℃, respectively.".

Point 11: Line 121, Change “The each sample was…” to “ Each of the samples was…”

Response 11: Thank you for your careful review. The sections of lines number 121 have been rewritten after carefully amendment, which have been revised to  "Each of the samples was re-equilibrated to RT,".

Point 12: Line 185 - 186, change “…threaten caused by the OTA …” to “…risk caused by OTA …”

Response 12: Thank you for your careful review. The sections of lines number 185 to 186 have been rewritten after carefully amendment, which have been revised to  "which was that the method could effectively prevent the health risk caused by OTA to the operator during the testing process.".

Point 13: Line 255 correct “… HLPC method” to “…HPLC method”.

Response 13: Thank you for your careful review. The sections of lines number 255 have been rewritten after carefully amendment, which have been revised to  "the ten randomly selected corn samples naturally contaminated OTA was analyzed by ELISA and compared with the results of the HPLC method.".

Reviewer 3 Report

Manuscript titled Development of an anti-idiotypic VHH antibody and toxin free enzyme immunoassay for ochratoxin A in cereals is well written and designed. However this article some shortcoming to make it better for the readers.

1.     Authors clearly shown the advantage of anti-idiotypic VHH antibody for enzyme immunoassay for ochratoxin A and comparing with HPLC is a really good experiment.

2.     Authors should show the HPLC profile for figure 6.

3.     All the figures do not have the error bars. This being an analytical tool for ochratoxin A analysis authors should include error bars for each figure.

4.     Authors showed thermal stability by heating for 5 min in figure 4A and heated for various time points in Figure 4B. To support this experiment measuring the thermal denaturation curve using circular dichroism or fluorescence is required. This measurement provides clear transition midpoint for stability.

5.     More discussion warranted on disulfide bonds stabilizing VHH structure.

Author Response

Response to Reviewer 3 Comments

Point 1: Authors should show the HPLC profile for figure 6.

Response 1: Thank you for your careful review. The purpose of this experiment is to verify the accuracy of the newly established ELISA method by comparing the results of the method with the results of the HPLC method, and the previously published related papers directly draw the relationship curve for the two methods to make the result analysis more intuitive. Therefore, after repeated considerations, we decided not to show the HPLC profile to avoid cumbersome.

Point 2: All the figures do not have the error bars. This being an analytical tool for ochratoxin A analysis authors should include error bars for each figure.

Response 2: Thank you for your careful review. After modification, the figures 2 to 4 have been included error bars. However, figure 5 is obtained from two standard curves, so there is no way to set the error line. According to previous research papers, the correlation curves of ELISA and HPLC results of mycotoxins in contaminated samples were not set with error bars.

Point 3: Authors showed thermal stability by heating for 5 min in figure 4A and heated for various time points in Figure 4B. To support this experiment measuring the thermal denaturation curve using circular dichroism or fluorescence is required. This measurement provides clear transition midpoint for stability.

Response 3: Thank you for your careful review. In this experiment, VHH and monoclonal antibodies were treated at different temperatures and incubation times. After cooling to room temperature, the treated antibodies were tested for specific antigen binding activity by ELIAS to determine the thermal stability of the Nanobody. This method is widely used for the determination of nanobody thermostability. Circular dichroism is mainly used to determine the steric structure of proteins, so later our laboratory will specifically study the structural mechanism to explain why VHH 2-24 is more stable than monoclonal antibodies.

Point 4: More discussion warranted on disulfide bonds stabilizing VHH structure.

Response 4: Thank you for your careful review. The sections of lines number 126-131 have been rewritten after carefully amendment, which have been revised to "This result was expected because VHH can form an additional disulfide bond between CDR3 and CDR1 or FR2 in addition to the conserved disulfide bond within the domain. Hence, the increased sequence of CDR1 and CDR3 and loop structure of VHH expands the area of antibody binding to antigen and the diversity of antibodies, while increasing the stability of its structure, resulting in VHH that can withstand high temperatures and harsh extreme environments.".
